# High-resolution, high-throughput detection of hidden antibiotic resistance with the dilution-and-delay (DnD) susceptibility assay

Muqing Ma [1,2] & Minsu Kim [1,2,3] ✉

Rising rates of antibiotic treatment failure highlight the complexity of resistance mechanisms. While genetically encoded resistance is well established, recent clinical studies have uncovered noncanonical mechanisms driven by phenotypic heterogeneity, such as heteroresistance, persistence, or adaptive resistance. Yet, standard susceptibility tests lack the resolution or throughput to detect these diverse phenotypic mechanisms. To address these limitations, we develop a scalable, high-resolution assay—Dilution-and-Delay (DnD)—by implementing two basic principles of bacterial growth. DnD detects rare drug-insensitive cells at frequencies as low as 1 in 100 million, while also reporting bulk population inhibition as the conventional MIC. We demonstrate this capability across synthetic communities, heteroresistance, persistence, and evolutionary progression. These high-resolution data expose how traditional susceptibility assays are constrained by statistical detection limits. Scaling up DnD for high-throughput application to ~120 clinical isolates reveals that under-the-radar multidrug resistance or tolerance is common. By combining resolution with scalability, DnD provides an advanced platform for antibiotic susceptibility testing with broad impact on basic research, clinical practice, and epidemiological surveillance. It supports a new framework for measuring, defining, and understanding antibiotic resistance.

Antibiotics are foundational to modern medicine—significantly improving quality of life, extending human lifespans, and enabling complex medical procedures. However, treatment failure due to inappropriate antibiotic use remains common, worsening patient outcomes and increasing the burden on healthcare systems[1,2]. Antibiotic susceptibility testing plays a critical role in combating this threat by guiding clinical decisions and ensuring treatments are tailored to the infecting pathogen.

Concerns about antibiotic failure date back to the very beginning of the antibiotic era, when Alexander Fleming warned that improper use of penicillin could drive the emergence of resistance—a prediction that proved prescient. Conventional antibiotic resistance arises by the acquisition of genetically encoded determinants

through mutations or horizontal gene transfer[3], which confer population-scale advantages under antibiotic pressure.

However, recent studies have shed new light on diverse mechanisms by which bacteria evade antibiotic treatments. Bacteria can survive through tolerance, where cells withstand drug exposure by decreasing their growth or entering a dormant state[4,5]. Importantly, both resistance and tolerance can manifest heterogeneously within populations. In heteroresistance, only a minority exhibits resistance in a population, while the majority of cells remain susceptible[6,7]. In persistence, a subpopulation transiently enters an antibiotic-tolerant state, surviving lethal antibiotic exposure and resuming growth once treatment ends[8,9]. Adaptive resistance, induced by sublethal antibiotic exposure, can further enable small subpopulations to gradually

[1]Department of Physics, Emory University, Atlanta, GA, USA. [2]Antibiotic Resistance Center, Emory University, Atlanta, GA, USA. [3]Graduate Division of Biological and Biomedical Sciences, Emory University, Atlanta, GA, USA. ✉e-mail: minsu.kim@emory.edu

withstand higher drug concentrations[10,11]. These traits can propagate through lineages and be stably maintained in the population[12,13].

Importantly, these resistant or tolerant subpopulations (hereafter referred to as antibiotic-insensitive) are often part of a broader evolutionary landscape. Heteroresistance or persistence often emerges early during the evolution under antibiotic treatments[14–16]. Once such subpopulations are present, inappropriate treatments can accelerate their acquisition of genetic resistance—providing an alternate accelerated route to population-wide resistance[14–17].

However, the true magnitude of this problem remains elusive because it routinely goes undetected in the clinic and research labs[18,19]. A major obstacle is the limited resolution of conventional antibiotic susceptibility testing. The current gold-standard assay measures population growth in antibiotic-containing microbroth, reporting the minimum inhibitory concentration (MIC)[20,21]—a single numerical value that summarizes the response of the entire population[22]. Disk diffusion and E-tests are alternative methods[23]. Disk diffusion measures inhibition zone diameters around antibiotic-impregnated disks on an agar plate, interpreting the MIC using zone diameter breakpoints. E-tests employ antibiotic gradient strips to provide an MIC estimate based on the point of growth inhibition. In all cases, the MIC obtained is compared with the pre-defined clinical breakpoint to classify the population as susceptible or resistant[24].

While simple and widely adopted, these current tests average over the population, masking underlying heterogeneity. As a result, they fail to inform non-canonical antibiotic insensitivity embedded within a susceptible population[25–27]. This, in turn, results in inappropriate therapy and "unexplained" treatment failures. More concerningly, when inappropriate therapy drives the evolutionary progression of minority variants to full-scale resistance[14–17], current tests remain blind to this process—detecting it only after resistant clones have swept through and become fixed in the population, by which point it is too late to adjust therapy and reverse the course. These limitations underscore the urgent need for high-resolution susceptibility testing capable of detecting antibiotic-insensitive cells even when they occur at very low frequencies within susceptible populations.

The ideal solution would be single-cell-level detection of bacterial survival. Leveraging advances in microscopy and microfluidics, we and others have measured antibiotic susceptibility at single-cell resolution, detecting survival frequencies as low as 1 in 10,000 ($10^{-4}$)[28,29]. While promising, this range is not sufficient to reliably detect non-canonical mechanisms of antibiotic insensitivity such as heteroresistance or persistence, which can occur at frequencies far below this range (e.g., $10^{-7}$ for heteroresistance)[25–27,30,31]. Moreover, these approaches depend on specialized instrumentation and technical expertise, making them inaccessible to most laboratories.

Historically, agar-based population analysis profiling (PAP) has been the method of choice for detecting heteroresistance[25–27]. In PAP, bacterial cultures are plated across a range of antibiotic concentrations, and colonies that grow at high concentrations are counted to quantify resistant cells. Although highly sensitive, PAP is labor-intensive and time-consuming—requiring agar preparation and manual CFU counting—making it impractical for routine diagnostics. This diagnostic blind spot obscures the diverse mechanisms underlying treatment failure and evidence-based clinical guidelines.

To address this challenge, we investigated two fundamental principles of bacterial growth, dilution-to-extinction and delay-to-growth, thereby developing a practical, scalable, high-resolution strategy for antibiotic susceptibility testing. This assay retains the operational efficiency of conventional susceptibility tests, making it readily adoptable. Yet, it detects rare antibiotic-insensitive cells at frequencies as low as 1 in 100 million, while simultaneously reporting the bulk response as the standard MIC. We demonstrated its utility across synthetic communities, heterogeneous strains, and persistent populations, and by tracking the evolutionary dynamics early from the onset. Scaling it to high throughput, we uncovered a widespread prevalence of multidrug insensitivity in clinical isolates. Our analysis also shows that in heterogeneous populations, the MIC largely reflects the statistical sampling limits of conventional assays rather than a true population-wide inhibitory dose. These findings establish antibiotic susceptibility as a quantitative spectrum rather than a binary trait, providing a new framework for measuring, defining, and understanding bacterial response to antibiotics.

## Results

The gold standard for antibiotic susceptibility testing is broth-based culture, standardized by international organizations, including the Clinical and Laboratory Standards Institute (CLSI)[20] or the European Committee on Antimicrobial Susceptibility Testing (EUCAST)[21]. These protocols specify inoculating a fixed number of bacterial cells ($N$, typically ~$5 \times 10^5$) into antibiotic-containing media and visually observing the turbidity after a defined period ($T$, typically ~18 h) to determine the MIC of the sample. However, $N$ and $T$ are procedural parameters—not intrinsic biological properties. We hypothesized that systematically varying these parameters could reveal resistant cells hidden in a susceptible population.

### Dilution-to-extinction quantifies rare resistant cells through serial dilution

Consider a bacterial culture in which resistant cells occur at a frequency of $10^{-7}$. If the standard inoculum $N \approx 5 \times 10^5$ is used, it will likely contain no resistant cells and will be sterilized by antibiotic treatment. In contrast, a culture of $N \approx 10^9$ cells would almost certainly include resistant cells, which will grow in the presence of antibiotics to turn the culture turbid. Therefore, by systematically varying $N$, we can infer the abundance of resistant subpopulations based on turbidity.

We formalized this concept into a dilution-to-extinction assay (Fig. 1a). Starting from a high-density culture, we performed serial dilutions and incubated them in antibiotic-containing media. At low dilutions, resistant cells are likely present and grow, driving turbidity. As the dilution increases, the probability of including resistant cells decreases, and eventually, a dilution contains none—resulting in no growth. The first dilution in the series that lacks turbidity defines the extinction point (Fig. 1a). For example, if the original population contains a few tens of resistant cells, the first 10-fold dilution would still turn turbid, while the second would not. In our analysis, we determined the number of dilutions required to reach the extinction point to estimate the number of resistant cells in the original sample.

To validate this approach, we spiked a susceptible strain with known fractions of a fully resistant strain, preparing defined mixtures. We applied the dilution-to-extinction protocol in the presence of antibiotics to estimate resistant cells, and in parallel, performed the same dilution series without antibiotics to estimate the total number of viable cells (Fig. 1a). Comparing the extinction points across the two conditions enabled us to compute the resistance frequency (RF) in each mixture (Fig. 1a).

Across samples with resistant fractions spanning several orders of magnitude, the measured RF values closely recapitulated the known input frequencies (Fig. 1b). Replicate measurements, indicated by three different symbols, exhibited ~10-fold variation, consistent with the resolution limit imposed by 10-fold serial dilution. Nonetheless, the data showed a strong linear relationship, with a slope of 0.95 and an $R^2$ of 0.98 (Fig. 1b). Notably, the method reliably detected resistant cells at frequencies as low as ~$10^{-8}$— below a conventional heteroresistance threshold of ~$10^{-7}$[27]. This test indicates that the dilution-to-extinction assay enables unbiased detection of resistant subpopulations across a broad dynamic range.

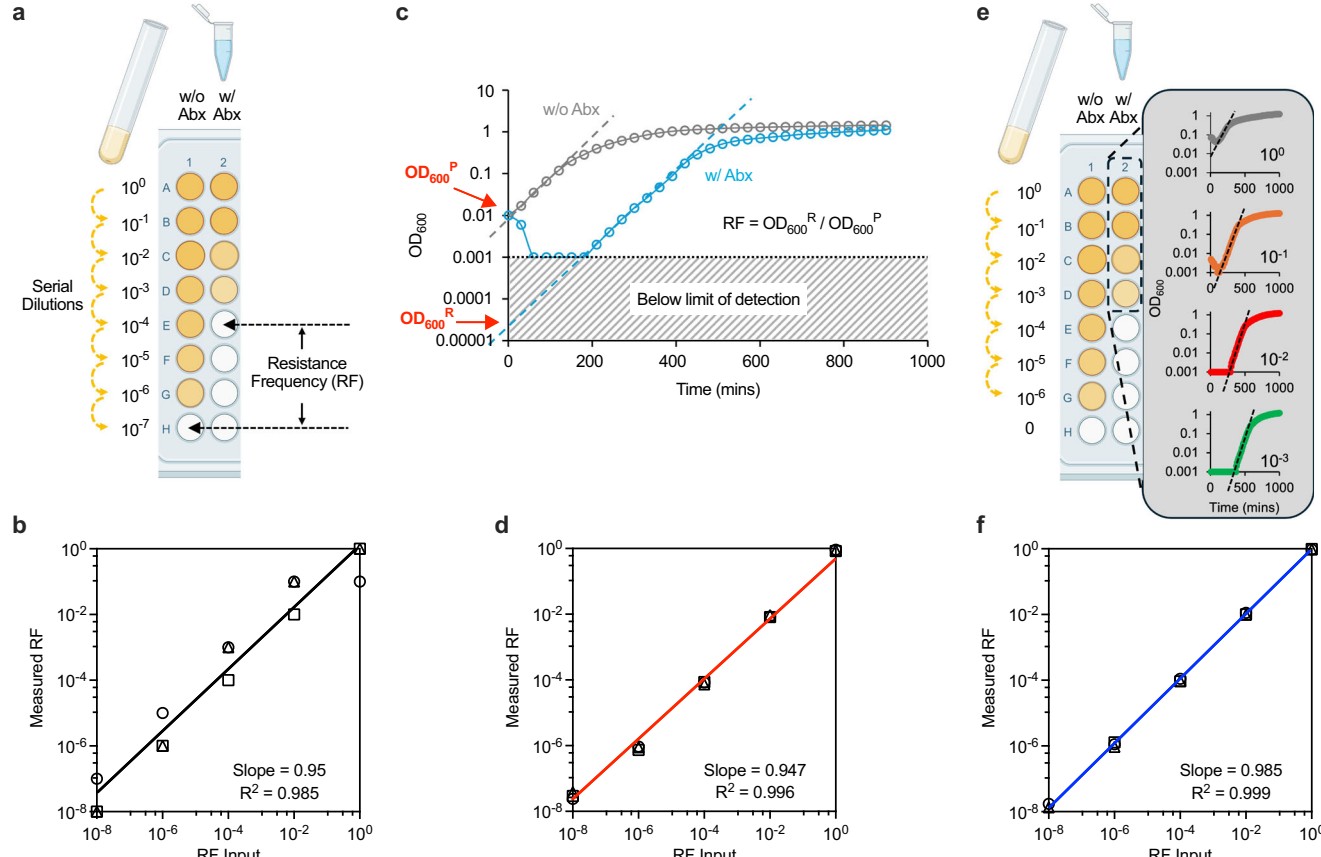

**Fig. 1 | High-resolution quantification of rare antibiotic-resistant subpopulations. a** Dilution-to-extinction assay. A bacterial culture ($10^9$ cells/mL) is serially diluted and incubated with and without antibiotics (abx). The first dilution lacking visible turbidity defines the extinction point, which estimates the number of resistant cells in the original population. **b** To validate this approach, we identified a colistin-susceptible (AMK105) and a fully resistant (AMK104) strain by PAP; i.e., resistance frequency (RF) = 0 and 1, respectively, measured at the clinical break point (4 µg/mL). We mixed the two strains at defined ratios, inferred RF for each mixture using dilution-to-extinction, and compared the inferred values to the input ratios. The inferred RFs closely matched the inputs across several orders of magnitude. Linear regression yielded a slope near 1 and $R^2 = 0.985$. **c** Delay-to-growth assay. In antibiotic-treated cultures, the initial $OD_{600}$ decline reflects killing of the susceptible population, followed by delayed regrowth from rare resistant cells. Exponential regrowth is extrapolated back to time zero to estimate $OD_{600}^R$, representing the density of resistant cells in the original culture. Comparing this to

the initial $OD_{600}^P$ of the total population yields an estimate of RF. **d** RF values inferred from delay-to-growth closely matched the known input frequencies, with strong linear correlation (slope ≈ 1, $R^2 = 0.996$). **e** Dilution-and-Delay (DnD) assay. The two methods were integrated by performing serial dilutions and continuously monitoring $OD_{600}$ in each well. Wells without growth were used to define the extinction point (Fig. 1a), while wells that exhibited regrowth were analyzed to infer their individual RF values (Fig. 1c); these values were averaged to yield the final RF. This RF value was then compared with the estimate from the extinction points for cross-validation (Supplementary Fig. 1). **f** The RF value obtained from DnD was plotted against known input frequencies, showing strong quantitative agreement (slope ≈ 1) with the highest precision ($R^2 = 0.999$) among the three approaches. Each experiment was performed in biological triplicate. Individual replicates are shown as different symbols, with lines and error bars indicating the mean and standard deviation. Created in BioRender. Ma, M. (2026) https://BioRender.com/i14aaw4.

## Delay-to-growth captures rare survivors through density monitoring

The other fixed parameter in standard antibiotic susceptibility testing is the observation time ($T$), typically ~18 hours[20,21]. This constraint assumes that antibiotic-sensitive and -resistant populations will fully diverge within that period. However, when resistant cells are rare, their outgrowth is delayed. The less resistant cells are present, the longer this delay. The core idea behind the delay-to-growth assay is to analyze this delay under antibiotic treatment to infer the abundance of resistant cells.

We tested this principle using the defined mixtures of susceptible and resistant strains. Optical density ($OD_{600}$) was measured over time using a spectrophotometer to monitor culture growth. In the absence of antibiotics, the culture exhibited immediate exponential growth without lag (Fig. 1c, grey). In contrast, when antibiotics are present, the $OD_{600}$ initially declined due to killing of susceptible cells—often dropping below the detection limit of a spectrophotometer (Fig. 1c,

cyan). This was followed by delayed increase in $OD_{600}$, corresponding to the outgrowth of surviving resistant cells that become detectable.

To estimate the number of resistant cells in the original culture, we fit the exponential phase of the recovery curve, extrapolating it back to time zero (dashed line, Fig. 1c). This yielded an $OD_{600}$ of resistant subpopulations at $t = 0$, denoted $OD_{600}^R$ (Fig. 1c). To calculate its frequency relative to the total population, we compared it with the total $OD_{600}$ of the culture ($OD_{600}^P$ in Fig. 1c). The resulting ratio, $OD_{600}^R/OD_{600}^P$, provides an estimate of RF in the original sample.

To assess the quantitative accuracy and dynamic range of the method, we varied the abundance of resistant cells in the defined mixture over several orders of magnitude and measured the RFs. The measured RF values closely matched the known input ratio, yielding a linear regression with a slope of 0.95 with $R^2$ of 0.996 (Fig. 1d). Therefore, the delay-to-growth approach accurately captures resistant subpopulations.

## Integrating two axes of resolution: the dilution-and-delay (DnD) framework

To improve both quantitative accuracy and operational efficiency, we integrated the dilution-to-extinction and delay-to-growth assays into a unified framework. Since both assays are performed in broth and rely on turbidity as the readout—whether visual inspection or optical density reading—they are naturally compatible. Moreover, turbidity can be monitored automatically via $OD_{600}$ measurements using standard plate readers, enabling seamless integration into high-throughput workflows. Importantly, the two assays are based on distinct underlying principles, yielding orthogonal estimates of RF. Cross-validating these independent measurements provides an internal consistency check, thereby increasing the robustness and reliability of RF quantification.

We implemented this combined strategy as the Dilution-and-Delay (DnD) assay. Following the dilution-to-extinction protocol, we prepared serial dilutions in a 96-well microtiter plate, then continuously monitored $OD_{600}$ in each well over time using a plate reader (Fig. 1e). Wells that failed to show an $OD_{600}$ increase were used to define the extinction point as described in the dilution-to-extinction method (Fig. 1a). The wells that exhibited growth were used to estimate RF values, based on the delay-to-growth method (Fig. 1c). The dilution series also served as technical replicates (Fig. 1e); RF values from individual wells were averaged to improve statistical precision, yielding the final RF for that experiment.

We validated the DnD assay using defined mixtures of fully resistant and susceptible strains. Across all input ratios, the averaged RF was within the ~10-fold uncertainty inherent to the dilution-to-extinction method alone, showing an internal check of consistency (Supplementary Fig. 1). Linear regression of input versus the RFs showed a slope closest to 1 and the highest $R^2$ (0.999), exhibiting near-perfect agreement (Fig. 1f).

## Validation against heteroresistant clinical isolates

We next applied our assays to a panel of five clinical isolates previously characterized as heteroresistant against various antibiotics[17,18,32]. Unlike the defined artificial mixtures used above, these strains endogenously harbor resistant subpopulations with variable frequencies and resistance levels. Each strain had been previously evaluated using population analysis profiling (PAP)—a well-established method for identifying heteroresistance—revealing a gradual decline in CFUs with increasing antibiotic concentrations (red symbols in Fig. 2a–c for three strains; Supplementary Fig. 2 for the remaining two strains). This gradual decrease reflects a heterogeneous distribution of resistance within each population. By comparing CFUs at each antibiotic concentration to those in antibiotic-free conditions, we obtained $RF_{PAP}$ (red in Fig. 2). Three replicate measurements were averaged to calculate the mean $RF_{PAP}$: $\langle RF \rangle_{PAP}$.

Although PAP is accurate, its labor- and time-intensive nature limits its utility in clinical and high-throughput settings. We therefore evaluated our DnD (green), dilution-to-extinction (grey), and delay-to-growth (yellow color) assays on the same strains. Measurements were repeated three times for each condition. All three assays produced RFs that closely matched one another and recapitulated the gradual decline observed in PAP (Fig. 2a–c and Supplementary Fig. 2).

For quantitative comparison, we characterized how our assay results deviate from those of the PAP assay by normalizing their RFs by $\langle RF \rangle_{PAP}$, thereby quantifying their deviation by $RF/\langle RF \rangle_{PAP}$ for each strain and antibiotic concentration. Aggregating these values (open symbols, Fig. 2d) revealed that RF values from each of our assays were centered around the $\langle RF \rangle_{PAP}$, indicating their strong agreement. Dilution-to-extinction exhibited the most variance, about 10-fold, as expected. The DnD assay exhibited the lowest variance.

We further performed statistical comparison using a one-way ANOVA test between RF values from each assay and those from PAP. All

tests yielded $p > 0.05$, indicating that our assay results and PAP results are not statistically different (Fig. 2d). This finding aligns with earlier comparisons using defined mixtures, where each assay accurately captured the input frequency (Fig. 1). Notably, among the three assays, the DnD assay produced the highest $p$-value (approaching 1), suggesting its measurements are quantitatively indistinguishable from PAP. Together, these tests demonstrate that all three assays faithfully quantify antibiotic-resistant subpopulations, with the DnD assay providing the greatest accuracy and reproducibility.

## The DnD assay reveals that the sampling limit determines MIC

The MIC is widely used as a measure of antibiotic resistance, traditionally defined as the lowest drug concentration that prevents population-level growth[22]. While the DnD assay was designed to quantify rare antibiotic-insensitive subpopulations, its serial dilution format includes wells containing the conventional inoculum size ($\sim 5 \times 10^5$ cells[20,21]), enabling simultaneous determination of MIC by standard criteria.

To clarify the meaning of MIC in heterogeneous populations, we measured the MIC of artificial mixtures containing defined ratios of fully susceptible and fully resistant cells (Fig. 1). At low resistant frequencies, the MIC matched that of the susceptible strain (1 µg/mL), whereas at high frequencies it matched that of the resistant strain (1024 µg/mL); see Supplementary Fig. 3. A sharp transition occurred when the resistant frequency crossed $\sim 3 \times 10^{-6}$ (vertical dashed line). Following conventional classification based on the breakpoint (4 µg/mL, horizontal red line), the population would therefore flip from "susceptible" to "resistant" at this threshold, illustrating the flaws of binary interpretation in heterogeneous populations.

This threshold was also evident in endogenously heteroresistant strains (Fig. 2). We determined the MIC of these strains by visually inspecting wells inoculated with the standard inoculum size (Supplementary Fig. 4a). We then plotted the resistance frequency (RF) at the MIC for each strain (Supplementary Fig. 4b). Across all strains, the MIC values corresponded to the first antibiotic concentration at which RF dropped below $\sim 3 \times 10^{-6}$ (dashed line). This threshold can be understood based on the standard inoculum size ($\sim 5 \times 10^5$ cells); at this inoculum size, a population with an RF below $\sim 3 \times 10^{-6}$ is unlikely to contain resistant cells, resulting in complete clearance and thus reporting the MIC.

These results demonstrate that MIC marks the point at which resistant survivors fall below detection, representing the statistical sampling limit of conventional assays. Consequently, resistant subpopulations below this threshold escape detection, underscoring the need for high-resolution approaches such as the DnD assay.

## The DnD enables detecting early stages of evolution and its dynamics

The evolution of antibiotic resistance is often tracked by monitoring changes in MIC[33–35]. Our above analysis reveals that such changes become apparent only when resistant subpopulations expand beyond the sampling threshold of conventional assays, leaving the earliest stages of reduced susceptibility hidden. However, early detection is critical both clinically, where interventions are more effective before resistant mutants become established[35,36], and scientifically, where direct observation can shed light into the selective dynamics driving their establishment[36]. With its ability to quantify resistant minorities with high resolution, the DnD is ideally suited to track the early stages of resistance evolution.

To demonstrate this, we evolved a susceptible *E. cloacae* strain under colistin exposure by serially passaging the culture for several days. The DnD assay captured the initial appearance of resistant mutants, as well as their gradual expansion over time (Supplementary Fig. 5a). Only when this subpopulation crossed the sampling threshold did it trigger a measurable increase in MIC (Supplementary Fig. 5b).

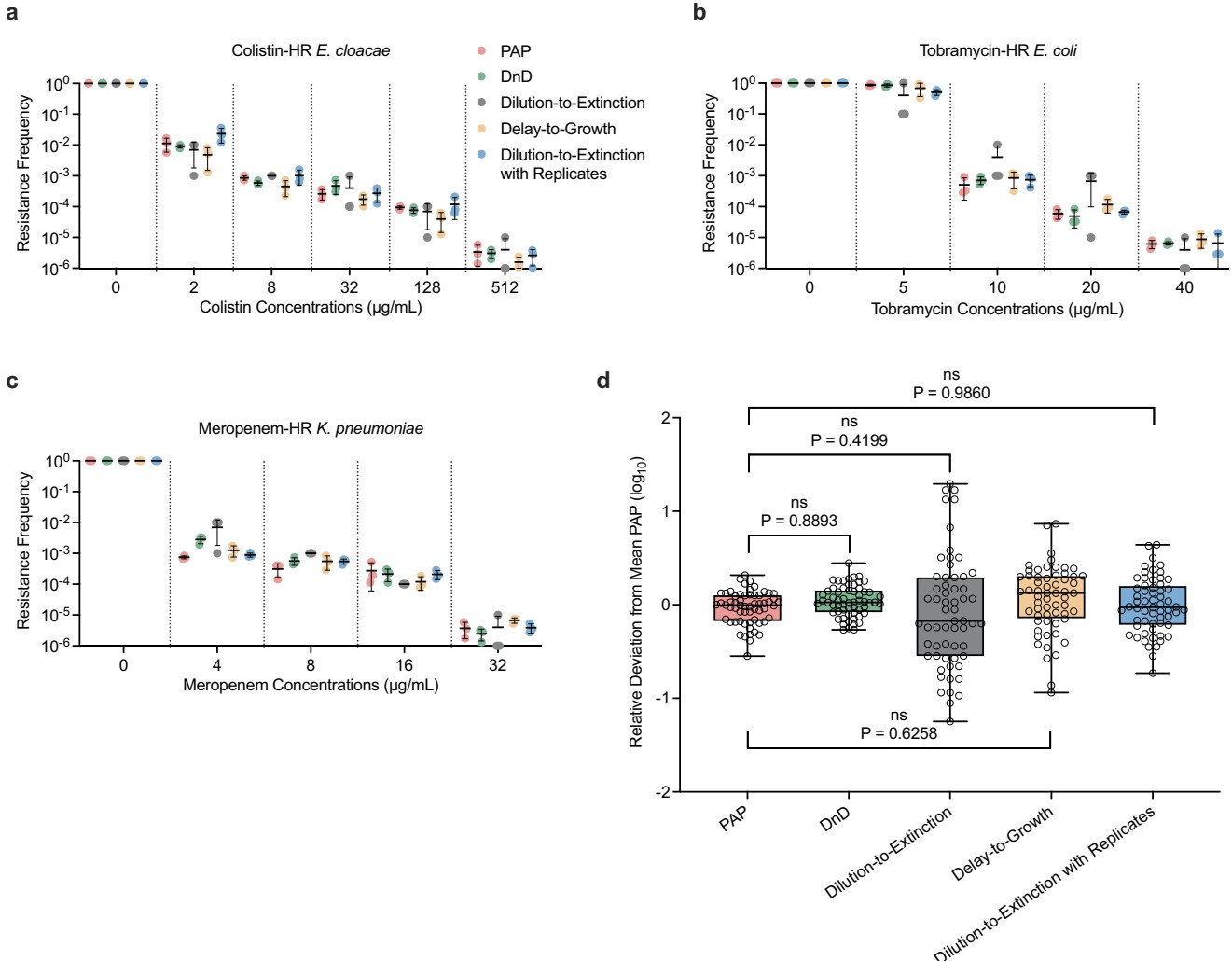

**Fig. 2 | Validation of the DnD assay against heteroresistant clinical isolates.**
**a**–**c** Resistance frequencies (RFs) were measured for five clinical heteroresistant (HR) isolates: colistin-HR *E. cloacae* (AMK107), tobramycin-HR *E. coli* (AMK117), and meropenem-HR *K. pneumoniae* (KMK4). See Supplementary Fig. 2 for two additional HR strains (AMK118, AMK120). In prior studies, population analysis profiling (PAP) was used to determine the fraction of resistant subpopulations. For direct comparison, we applied the DnD assay alongside dilution-to-extinction and delay-to-growth assays. Each assay was performed in biological triplicate; individual replicates are shown as different symbols, with black lines and error bars indicating the mean and standard deviation. All three assays recapitulated PAP-derived RF, $RF_{pap}$, across antibiotics and concentrations. **d** RF values from all assays (DnD,

dilution-to-extinction, delay-to-growth) were statistically compared against $RF_{PAP}$. Data from (**a**–**c**) and Supplementary Fig. 2 were pooled, and each RF value was normalized to the corresponding mean PAP value (<RF>$_{PAP}$) to calculate relative deviation on a $\log_{10}$ scale. The distribution of $\log_{10}(RF/(<RF>_{PAP}))$ is centered around 0, indicating strong agreement across methods. Box plots show the median (center line), interquartile range (box), and minimum to maximum values (whiskers), with all individual data points shown ($n = 60$). One-way ANOVA showed no statistically significant differences between methods ($p > 0.05$), denoted as "ns." Notably, the DnD assay exhibited the lowest variance and a high $p$-value, underscoring its superior accuracy and reproducibility.

This experiment demonstrates that DnD can identify the emergence of resistance early in evolution.

## High-throughput screening of clinical isolates uncovers widespread multi-drug heteroresistance

Population heterogeneity—particularly the presence of minority resistant cells within seemingly susceptible populations—challenges binary classification. To clarify how antibiotic susceptibility should be measured and quantified, we next examined how often clinical isolates harbor such minorities and how broadly their resistance frequencies (RF) are distributed.

In addition to quantitative accuracy, another key advantage of the DnD assay is its compatibility with automated, high-throughput screening. Because the assay is implemented in microtiter plates and utilizes optical density as a readout, it can be fully automated using a standard plate reader. We applied the DnD assay to a panel of ~120

previously uncharacterized clinical isolates of *Klebsiella pneumoniae*, *Acinetobacter baumannii*, *Pseudomonas aeruginosa*, *Enterobacter cloacae*, and *Escherichia coli*. These strains were obtained from the Multi-site Gram-negative Surveillance Initiative (MuGSI), CDC. Each isolate was tested against five antibiotics: meropenem and piperacillin/tazobactam (β-lactams), tobramycin (aminoglycosides), ciprofloxacin (fluoroquinolones), and colistin (polymyxins).

For each strain and drug, we determined the RF at the clinical breakpoint concentration. Comparison with independently performed PAP assays showed strong quantitative agreement: $RF_{DnD}$ versus $RF_{PAP}$ yielded a linear relationship with slope = 0.95 and $R^2 = 0.95$ (Fig. 3a and Supplementary Fig. 6 shows individual antibiotics).

We next plotted the histogram of $RF_{DnD}$, which revealed a trimodal distribution (Fig. 3b). Two sharp peaks on the left and right reflected conventional binary categories: susceptible (RF = 0) and

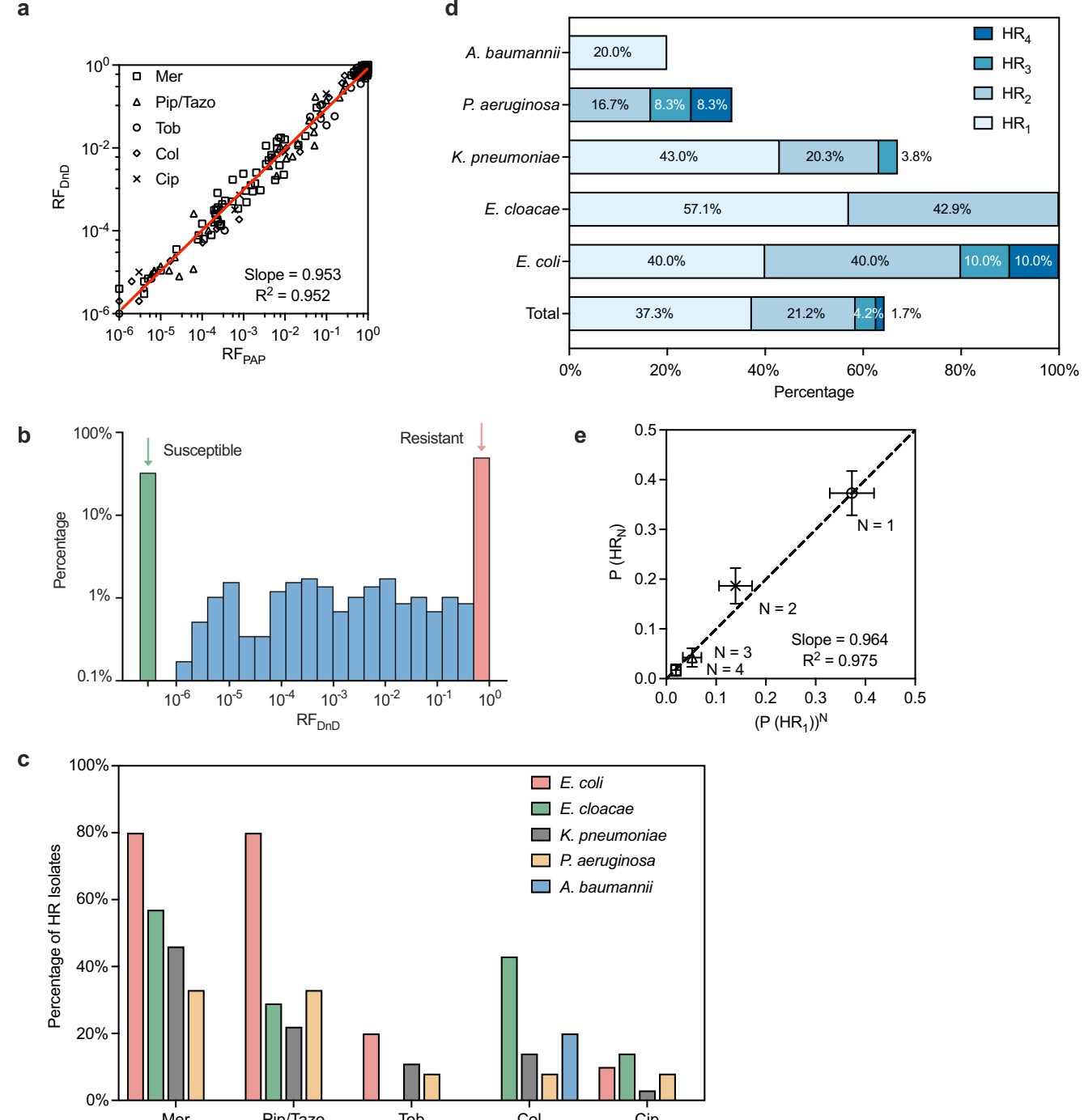

**Fig. 3 | High-throughput application of the DnD.** The DnD assay was applied to ~120 clinical isolates of *Klebsiella pneumoniae*, *Acinetobacter baumannii*, *Pseudomonas aeruginosa*, *Enterobacter cloacae*, and *Escherichia coli* at the clinical breakpoints of five antibiotics: meropenem (Mer), piperacillin/tazobactam (Pip/Tazo), tobramycin (Tob), colistin (Col), and ciprofloxacin (Cip). **a** PAP assays were conducted independently for comparison. $RF_{DnD}$ versus $RF_{PAP}$ yielded a linear relationship with slope = 0.953 and $R^2$ = 0.952, showing strong agreement. See Supplementary Fig. 6 for individual antibiotics. **b** Histogram of $RF_{DnD}$ revealed a trimodal distribution. Two peaks on the left and right reflected conventional binary categories: susceptible (green bar, RF = 0) and resistant (RF ≈ 1). Strikingly, a broad plateau between these extremes spans several orders of magnitude, containing 18.8% of test results. The absence of a single peak in this intermediate region highlights that resistance is not a binary trait, but instead varies continuously across a quantitative spectrum. resistant (red bar RF ≈ 1). A broad intermediate plateau encompassed 18.8% of test results and was designated as heteroresistance (HR) for additional analyses; see Supplementary Table 1. **c** HR prevalence varied across antibiotics, being most frequent for β-lactams. **d** HR across all species together (first column) and individually. Multi-drug HR was common in *E. coli* and *P. aeruginosa*. **e** Observed probabilities of multi-drug HR. We counted the strains exhibiting HR to n antibiotics per strain (*n* = 1, 2, 3, 4, or 5) and calculated the probability P(HRₙ). It is plotted against the expectations if HR to each antibiotic occurred independently ($P(HR_1)^2$, $P(HR_1)^3$, $P(HR_1)^4$). Error bars represent variability in the estimated probabilities.

resistant (RF ≈ 1). Strikingly, a broad plateau between these extremes spans several orders of magnitude, containing 18.8% of test results. The absence of a single peak in this intermediate region highlights that resistance is not a binary trait, but instead varies continuously across a quantitative spectrum.

Next, we asked how different strains and antibiotics are represented within this intermediate region. For this analysis, we labeled this broad region collectively as heteroresistance (HR); see Supplementary Table 1. The prevalence of HR varied widely by both antibiotics (Fig. 3c) and bacterial species (Fig. 3d). For example, HR was frequent for β-

lactams (meropenem and piperacillin–tazobactam), but less common for tobramycin (aminoglycoside) and ciprofloxacin (fluoroquinolone). Notably, multi-drug HR was also observed: ~21% of isolates displayed HR to two antibiotics $P(HR_2)$, and ~6% to three or more $P(HR_3)$ and $P(HR_4)$. Multi-drug HR was particularly enriched in *E. coli* and *P. aeruginosa*.

An important question is whether heteroresistance to one antibiotic promotes heteroresistance to others, or whether they are independent. To address this, we measured the probabilities of multi-drug HR: $P(HR_2)$, $P(HR_3)$, and $P(HR_4)$. We then compared these values to the expectations if HR events occurred independently ($P(HR_1)^2$, $P(HR_1)^3$, $P(HR_1)^4$). Remarkably, the observed and expected values aligned along the y = x line (Fig. 3e). This alignment suggests that acquisitions of multi-drug HRs are independent events. However, we note that these estimates carry large error bars due to the limited sample size. Thus, additional studies with larger cohorts will be required to confirm this trend.

### The DnD assay reveals widespread persistence

Persistence refers to a phenomenon in which a subpopulation of bacterial cells enters a non-growing or abnormally slow-growing state, allowing them to tolerate otherwise lethal antibiotic treatment[31]. This tolerant state is transient: once the antibiotic is removed, these persister cells can exhibit normal growth.

Persistence is typically measured using a CFU-based time-kill assay, which exploits this transiency[31]. In this assay, a bacterial culture was exposed to high concentrations of a bactericidal antibiotic to eliminate actively growing cells. Samples are collected at different time points, washed, and plated on antibiotic-free agar; CFUs reflect the number of surviving persisters. However, like PAP tests, this approach requires manual plating and colony counting, making it labor-intensive and time-consuming.

We therefore evaluated whether our dilution-to-extinction, delay-to-growth, and DnD assays could also quantify persister cells. While these assays were previously performed in the presence of antibiotics to measure resistant cells that grow despite antibiotic exposure, persisters are not resistant and can grow only after the antibiotic is removed. For this reason, traditional persistence assays are conducted in antibiotic-free media following antibiotic exposure. We adopted the same strategy here, performing our assays in antibiotic-free broth after transient antibiotic treatment.

For rigorous tests, we first focused on well-characterized incidences. Previous work, including by our group, has shown that exponentially growing *E. coli* cultures contain negligible levels of persisters, whereas overnight stationary-phase cultures exhibit dramatically increased persistence[8,9]. Another model of enhanced persistence involves deletion of *bfmS* in *Acinetobacter baumannii*[37]. We applied our assays to both conditions, alongside CFU-based assays for comparison.

Across all cases, our assays accurately quantified persistence frequencies and reproduced the expected biological patterns (Fig. 4a-b). When we determine the deviation of each assay from the CFU-based assay (as done in Fig. 2f), we again found that DnD offers the greatest accuracy and reproducibility (Fig. 4c).

We then applied the DnD assay to a panel of ~120 clinical isolates to quantify their persistence frequencies (PF). We compared it with independently performed CFU assays: $PF_{DnD}$ versus $PF_{CFU}$ yielded a linear relationship with slope = 0.93 and $R^2$ = 0.98 (Fig. 4d), further demonstrating that the DnD can accurately quantify persistence. We next plotted the histogram of $PF_{DnD}$ (Fig. 4e). A large fraction of strains (38.1 %) contained persister cells, with their frequency varying over 5 orders of magnitude (red bars).

## Discussion

To address the complex and evolving landscape of antibiotic insensitivity, there is a pressing need for practical, high-resolution susceptibility testing that can be implemented across both research and clinical settings. The DnD assay was developed to meet this need. By leveraging two underutilized experimental axes—cell number and observation time—DnD enables accurate and quantitative detection of rare antibiotic-insensitive cells embedded within otherwise susceptible populations, down to frequencies of 1 in 100 million. This lower boundary is dictated by the practical sampling and observation designs, set by the maximum inoculum, dilution range, and observation window. It is low enough to capture the lowest-frequency range defined by operational categorization of heteroresistance[6,7].

Our results emphasize why high-resolution susceptibility measurements are essential. Conventional MIC-based susceptibility assays work well for detecting classical, genetically encoded resistance, where populations respond uniformly to antibiotics. However, in heterogeneous populations, our DnD measurements reveal that MIC largely reflects the statistical sampling limit of resistant minorities, rather than a true population-wide inhibitory dose. As a result, MIC-based binary classification is arbitrary, masking the presence of minority phenotypes and obscuring their clinical and evolutionary significance. High-resolution assays like DnD are therefore indispensable for capturing the full spectrum of antibiotic responses.

In the development of DnD, we have also shown that each of its axes—dilution-to-extinction and delay-to-growth—can be used independently, which already improves the detection limit significantly beyond the conventional assays. The dilution-to-extinction assay requires visual readouts of turbidity, making it suitable for resource-limited settings. Its reliance on 10-fold serial dilutions introduces an intrinsic ~10-fold uncertainty in RF estimates, as evidenced by a similar degree of deviation in our analysis. However, this level of precision is sufficient if the goal is to determine the orders of magnitude of resistant subpopulations. For greater quantitative resolution, the dilution factor can be reduced (e.g., from 10-fold to 2-fold), though this comes at the cost of exponentially increasing dilution steps.

A more practical strategy for improving precision of the dilution-to-extinction assay is to increase the number of biological replicates. In particular, the most probable number (MPN) analysis method provides a statistical framework for estimating the number of discrete units (e.g., resistant cells) based on the presence or absence of growth in multiple subdivided samples[38]. It was traditionally used to estimate bacterial abundance in environmental samples[39]. Here, we applied it to antibiotic-treated cultures to examine whether it can improve the RF estimate. We first tested how many (technical) replicates are needed to reliably measure RFs. We found that increasing the number of replicates indeed reduces the uncertainty in the RF estimate (Supplementary Fig. 7), although the benefit plateaus beyond five samples. We next applied this approach to the heteroresistant strains (Fig. 2). We prepared six replicates for each condition, conducted dilution-to-extinction analysis for each replicate, and applied the MPN, thereby determining the RF (Dilution-to-Extinction with Replicates, blue color in Fig. 2). We found that the RF estimates were comparable to those from PAP and DnD (Fig. 2d). Therefore, the inherent 10-fold uncertainty in the dilution-to-extinction method can be reduced using more samples, offering a useful strategy to improve quantitative accuracy in resource-limited settings without specialized equipment.

In contrast, the benefit of the delay-to-growth assay is that it requires only a single culture per condition. Therefore, it is ideal when sample size is limited. It relies on OD monitoring, which can be performed using a basic, widely available spectrophotometer costing less than $1000. Our data confirm that this method can detect rare resistant subpopulations. However, in standard laboratory settings, where these limitations are less restrictive, these two modular assays can be seamlessly integrated into the unified DnD framework to maximize detection resolution.

Although PAP can also provide high resolution, it is labor-intensive and time-consuming, requiring 2–3 h of hands-on time per

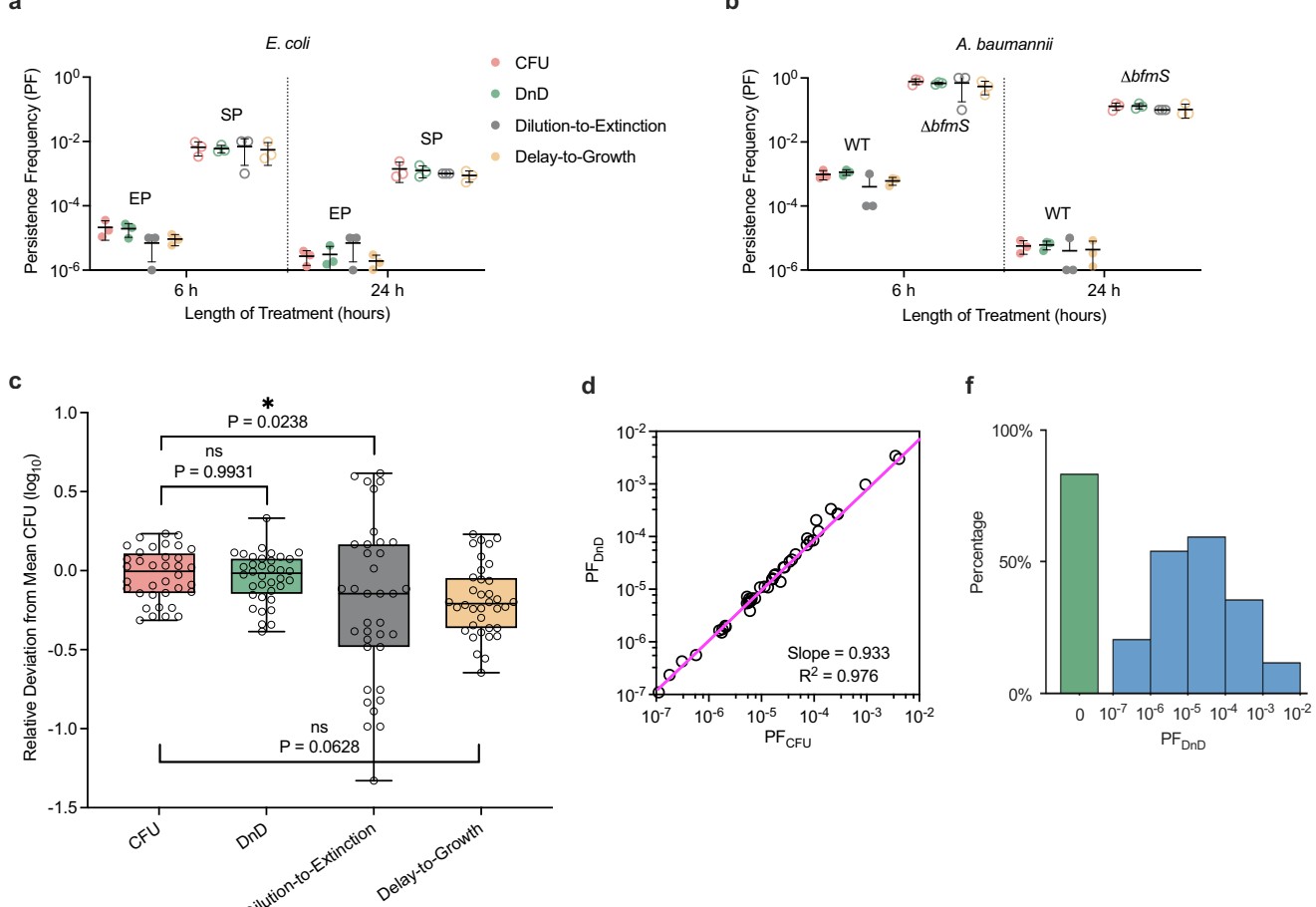

**Fig. 4 | Quantification of bacterial persistence using the DnD. a–c** Persistence frequencies (PFs) were measured at 6 and 24 hours after antibiotic treatment. Extended incubation in stationary phase in *E. coli*[9] and deletion of *bfmS* in *A. baumannii*[37] are known to elevate persistence levels. Traditionally, CFU-based assays have been used to quantify persistence. We quantified PFs using the DnD assay. Dilution-to-extinction and delay-to-growth were conducted for additional comparison. All methods recapitulated the expected biological trends and reproduced PFs obtained by CFU-based assays. Following published protocols, 100 μg/mL ampicillin for *E. coli*[9] and 128 μg/mL carbenicillin for *A. baumannii*[37,40] were used. EP: exponential phase; SP: stationary phase. Each assay was performed in biological triplicate; individual replicates are shown as different symbols, with black lines and error bars indicating the mean and standard deviation. In Fig. 4c, PFs determined by each method were statistically compared as described in Fig. 2d caption, which showed strong agreement across methods. Box plots show the median (center line), interquartile range (box), and minimum to maximum values (whiskers), with all individual data points shown (*n* = 36). **d** The DnD assay was applied to ~120 clinical isolates to quantify their PFs (see Fig. 3 caption for strain details). CFU assays were performed independently for comparison. $PF_{DnD}$ versus $PF_{CFU}$ showed strong agreement (slope = 0.933, $R^2$ = 0.976). **e** Histogram of $PF_{DnD}$ revealed that 38.1% of strains contained persister cells, with PF values spanning five orders of magnitude (blue bars).

condition, mostly devoted to agar preparation and manual CFU counting, which is unpractical for routine testing. In contrast, DnD uses a standard broth-based format and requires only 5–15 min of setup. Moreover, DnD is readily automatable: all measurements and analyses in this study were performed using a standard plate reader. The most labor-intensive steps—antibiotic media preparation and serial dilution—can be automated using increasingly available robotic liquid handling systems, further reducing the set-up time. Furthermore, when multiple antibiotics are tested, the primary scaling cost of DnD arises from increased well usage on microtiter plates. Because DnD operates at microliter-scale volumes, media and antibiotic consumption per condition is reduced by at least an order of magnitude relative to agar-based methods such as PAP, helping keep overall consumables costs low even for expanded testing (typically a few to low tens of dollars per isolate for a 15–20-drug panel).

DnD readouts could be influenced by several practical factors. First, post-antibiotic effects—including transient growth suppression after drug removal—can prolong time-to-detection, biasing delay-based back-extrapolation toward lower inferred survival frequencies.

Second, antibiotic instability (e.g., degradation, adsorption, or precipitation) can effectively reduce drug exposure over time and inflate apparent survival, biasing inferred frequencies upward. Third, atypical growth phenotypes, such as small-colony variants or severely slow-growing resistant variants, could further complicate inference from time-to-outgrowth. Importantly, these confounds reflect general limitations of phenotype-based susceptibility assays. In DnD, such effects are expected to produce discordance between dilution- and delay-based estimates. We therefore recommend cross-validating these estimates; the discordance should be treated as a diagnostic flag that motivates targeted follow-up (e.g., regrowth controls or growth-rate measurements). Importantly, in our experiments, we did not encounter discordance between dilution- and delay-based estimates, indicating that these confounds are unlikely to be common.

High-throughput application of DnD revealed that many clinical isolates harbor antibiotic-insensitive minority populations whose frequencies span several orders of magnitude. This finding again challenges the binary classification and reframes antibiotic susceptibility as a quantitative spectrum. The broad distribution of resistant minorities

likely reflects diverse underlying mechanisms and distinct evolutionary trajectories. More studies will be required to uncover the molecular origins, evolutionary progression, and true prevalence of these phenotypes. Importantly, our results establish DnD as an ideal platform for enabling such in-depth studies. For instance, combining DnD with genomic and transcriptomic profiling of heteroresistant subpopulations will enable direct linkage between resistance-frequency dynamics and the underlying molecular mechanisms driving their emergence and expansion.

We expect DnD to have positive clinical impact. Heteroresistance and persistence are often recognized retrospectively—when standard susceptibility testing indicates susceptibility, yet antibiotic therapy fails to clear infection. Early detection is therefore clinically valuable: identifying low-frequency survival phenotypes at the outset can inform antibiotic selection, dosing strategy, and monitoring before these subpopulations are selected and enriched during treatment. Our high-throughput application of DnD further suggests that the prevalence of heteroresistance and persistence may vary across species and antibiotic classes, motivating systematic studies to confirm these patterns. Establishing such organism–drug dependencies will help define the clinical contexts in which DnD is most informative.

While DnD enables high-throughput quantification of low-frequency survival phenotypes, we have not yet linked these measurements to outcomes in animal infection models or prospective patient cohorts. As a result, this study does not directly address how the assay would predict potential treatment failure or how the assay results should be interpreted to guide therapy or prognosis. At the same time, optimal clinical management of heteroresistant and persistent infections remains incompletely defined. Prospective clinical validation studies are critically needed to this gap. A practical barrier to validation is that current antimicrobial susceptibility testing is typically reported in a binary framework (susceptible/resistant) anchored to MIC-based breakpoints, whereas DnD returns a quantitative resistance frequency. Integrating such quantitative outputs into routine reporting will therefore require an interpretive framework that maps resistance frequency to actionable categories for therapy selection and stewardship. One practical approach may be to augment the current binary scheme with an additional category (e.g., heteroresistant), as has been proposed in prior work[6,7], together with standardized reporting thresholds defined in terms of resistance frequency and/or concentration range. The optimal strategy remains to be established and will require consensus standardization and outcome-linked validation. Nevertheless, reliable identification and quantification of low-frequency survival phenotypes is a necessary first step. DnD enables this at scale across pathogen–drug contexts to support future outcome-linked validation studies.

## Methods

### Bacterial strains and culture conditions

Colistin-susceptible and resistant *Enterobacter cloacae* (AMK105 and AMK104 in Fig. 1) were obtained from David Weiss' lab. Previously characterized heteroresistant strains (AMK107, AMK117, AMK118, AMK120, KMK4 in Fig. 2 and Supplementary Fig. 2)[17,18,32] were obtained from David Weiss' and Dan Andersson's lab. Clinical isolates (Supplementary Table 1) were obtained through the Georgia Emerging Infections Program, as part of the CDC's Multi-site Gram-negative Surveillance Initiative (MuGSI) in Georgia, USA. Wild-type *Acinetobacter baumannii* ATCC7978 and its Δ*bfmS* derivative (KMK111 and KMK112 in Fig. 4) were obtained from Isberg lab at Tufts University[37]. *Escherichia coli* K-12 strain NCM3722 (referred to as NMK1) was obtained from the Kim Lab bacterial collection[13].

Mueller-Hinton broth (MHB; BD Difco #275730) was used. Bacterial cultures were grown in MHB at 37 °C with shaking at 250 rpm, in 20 × 150 mm borosilicate glass culture tubes (Fisher Scientific #1496133). Exponentially growing cultures, typically at the $OD_{600}$ of 0.1, were used for the assay.

### Antibiotics

Colistin sulfate (#1264-72-8), tobramycin (#32986-56-4), gentamicin sulfate (#1405-41-0), and carbenicillin disodium (#4800-94-6) were obtained from Sigma-Aldrich. Meropenem trihydrate (#119478-56-7) was purchased from Research Product Industry. Tetracycline hydrochloride (#64-75-5) and ampicillin (#69-52-3) were obtained from Bio Basic.

### Population analysis profile (PAP)

PAP was performed as previously described[32]. Briefly, a single colony of a given strain, corresponding to a clonal population derived from one viable cell recovered from −80 °C glycerol stock, was inoculated into 2 mL MHB and grown with shaking at 250 rpm. After overnight growth (~16 h), cultures were diluted to $OD_{600} = 0.001$ and grown until $OD_{600}$ reached ~0.1 before being serially diluted in MHB in a 96-well plate (Corning #3598). From each dilution, 5 μL was plated on Mueller-Hinton agar (MHA; BD Difco #225250) containing antibiotics at indicated concentrations. Colonies were enumerated after 24 h. Resistance frequency (RF) at indicated antibiotic concentration was calculated by dividing the number of colonies on antibiotic-free MHA by the number of surviving colonies on MHA containing antibiotics.

### Dilution-to-extinction assay

For resistance frequency (RF) measurements, bacterial cultures were adjusted to approximately $10^9$ CFU/mL based on $OD_{600}$ readings and subjected to 10-fold serial dilutions in Mueller-Hinton Broth (MHB) up to $10^{-8}$. Dilutions were performed across rows in a 96-well microtiter plate (Corning #3596), with each well containing 200 μL of the diluted culture. For each dilution, parallel wells were prepared in MHB with or without indicated concentrations of antibiotics. Plates were incubated at 37 °C for ~18 h. After incubation, wells were visually inspected for turbidity. The first dilution in the series that lacks turbidity defines the extinction point. RF was estimated as the difference in dilution factor between the extinction points in antibiotic-free versus antibiotic-containing conditions.

For persistence frequency measurements, exponential-phase ($OD_{600}$ ~ 0.1) and 3 days stationary-phase ($OD_{600}$ ~ 4.0) cultures of *E. coli* NMK1, as well as exponential phase cultures of wild-type and Δ*bfmS A. baumannii*, were exposed to the indicated concentrations of antibiotics for 6 or 24 h. Following treatment, cultures were washed twice with phosphate-buffered saline (PBS) to remove residual antibiotics and then subjected to 10-fold serial dilutions in 200 μL MHB across rows of 96-well plates, as described above. Plates were incubated at 37 °C for ~24 h. After incubation, wells were visually inspected for turbidity. Persistence frequency was calculated by comparing the extinction point of the treated culture to that of the untreated control.

### Delay-to-growth assay

For resistance frequency measurements, exponential-phase bacterial cultures ($OD_{600}$ ~ 0.1) were inoculated into 200 μL Mueller-Hinton Broth (MHB) in a 96-well microtiter plate, with wells containing MHB with and without the indicated concentrations of antibiotics. Plates were incubated at 37 °C with shaking at 250 rpm for 18 to 24 h. $OD_{600}$ was measured and recorded at 10-min intervals using a spectrophotometer.

To estimate the number of resistant cells in the original population, the exponential phase of the recovery growth curve in antibiotic-containing wells was fitted. Exponential growth phases were identified based on both OD range and temporal continuity. Specifically, data points were required to fall within an OD600 range of 0.05–0.4, a regime in which optical density scales linearly with cell number under our experimental conditions. In addition, at least five consecutive time

points were required to determine the log-linear behavior. Linear regression was then performed on the semi-log-transformed data, and only fits with a coefficient of determination ($R^2$) $\geq 0.97$ were accepted for subsequent extrapolation and frequency estimation. The extrapolated OD600 at $t = 0$ ($OD_{600}^R$) represents the contribution of the resistant subpopulation. This value was normalized to the initial $OD_{600}$ of the total population ($OD_{600}^P$), yielding the resistance frequency (RF): $OD_{600}^R/OD_{600}^P$.

For persistence frequency measurements, *E. coli* NMK1 and *A. baumannii* (wild-type and Δ*bfmS*) were treated with the indicated concentrations of antibiotics for 6 or 24 h. After treatment, cells were washed twice with PBS to remove residual antibiotics, then re-inoculated into 200 μL of fresh MHB in a 96-well microtiter plate. Cultures were incubated at 37 °C with shaking at 250 rpm for ~24 h, with $OD_{600}$ monitored at 10-min intervals.

The $OD_{600}$ of the persister subpopulation ($OD_{600}^{PS}$) was estimated by extrapolating the exponential phase of the post-treatment growth curve back to time zero. Persistence frequency was calculated as the ratio of $OD_{600}^{PS}$ to the $OD_{600}$ of the total population ($OD_{600}^P$): $PF = OD_{600}^{PS}/OD_{600}^P$.

### Dilution-and-delay (DnD) assay

Exponential-phase ($OD_{600} \sim 0.1$) bacterial cultures were subjected to 10-fold serial dilutions in MHB across rows of a 96-well microtiter plate, with each well containing 200 μL of culture supplemented with or without the indicated concentrations of antibiotics. Plates were incubated at 37 °C with shaking at 250 rpm for approximately 18 h. $OD_{600}$ was recorded at 10-min intervals using the microplate reader.

For wells that exhibited detectable growth, the exponential phase of the recovery curve was fitted and extrapolated back to time zero as discussed above to determine RFs. The final RF value for a given replicate was obtained by averaging the individual RF values across all growing wells in the dilution series (See the flowchart in Supplementary Fig. 8 and the data analysis example in Source Data file).

For persistence frequency measurements, exponential-phase ($OD_{600} \sim 0.1$) and 3 days stationary-phase ($OD_{600} \sim 4$) cultures of *E. coli* NMK1, *A. baumannii* (wild-type and Δ*bfmS*), and ~120 clinical isolates ($OD_{600} \sim 0.1$) were exposed to the indicated concentrations of antibiotics for 6 or 24 h at 37 °C with shaking. After treatment, cultures were washed twice with fresh MHB to remove residual antibiotics, then serially diluted in 200 μL fresh MHB across rows of a 96-well plate as described above. Plates were incubated at 37 °C with shaking at 250 rpm, and $OD_{600}$ was recorded every 10 min for ~24 h.

### CFU-based time-kill assay

Exponential-phase ($OD_{600} \sim 0.1$) and 3 days stationary-phase ($OD_{600} \sim 4$) cultures of *E. coli* NMK1, *A. baumannii* (wild-type and Δ*bfmS*), and ~120 clinical isolates ($OD_{600} \sim 0.1$) were exposed to the indicated concentrations of antibiotics for 6 or 24 h at 37 °C with shaking. After treatment, 100 μL aliquots were removed from the cultures, washed twice with PBS to remove residual antibiotics, serially diluted in PBS, and plated on MHA. Plates were incubated at 37 °C for ~24 h, and colony-forming units (CFU) were enumerated to determine viable cell counts. To calculate persistence frequency, the number of surviving cells at 6 or 24 h was divided by the initial CFU count ($t = 0$) prior to antibiotic exposure.

### Evolution experiment under colistin exposure

To track early-stage evolutionary dynamics of antibiotic insensitivity, a colistin-susceptible *E. cloacae* strain (AMK105) was evolved under continuous colistin exposure by serial passaging. Unless otherwise noted, cultures were grown in MHB at 37 °C with shaking at 250 rpm.

At time 0, an AMK105 culture ($OD_{600} \approx 0.1$) was inoculated into fresh MHB containing colistin at 0.5 or 1 μg/mL (corresponding to 0.5× and 1× the ancestral MIC). Colistin concentration was increased stepwise over successive passages based on the emergence of resistant mutants in the evolving population, in order to maintain sustained selective pressure. Cultures were propagated for several days by serial passaging every 12 h, with each passage diluted to $OD_{600} \approx 0.1$ into fresh MHB containing the appropriate colistin concentration.

At indicated time points, aliquots were collected for phenotypic characterization. Resistance frequency (RF) at the selection concentration was quantified using the DnD assay as described above, and the emergence of resistant mutants was operationally defined as a sustained increase in RF relative to the ancestral population (Supplementary Fig. 5a). In parallel, MIC was measured at selected time points using a standard broth microdilution readout embedded within the DnD dilution series (wells inoculated at ~$5 \times 10^5$ cells), following CLSI/EUCAST criteria.

### Statistical analysis

Statistical analysis was conducted using GraphPad Prism version 10 (GraphPad Software). Details of biological replicates and statistical tests are provided in the corresponding figure legends. All data sets were tested for normality using the Shapiro-Wilk test and were confirmed to meet the normality criteria. Statistical analyses were performed as appropriate based on the experimental design.

### Reporting summary

Further information on research design is available in the Nature Portfolio Reporting Summary linked to this article.

## Data availability

Source data for figures are provided in the Source Data file. Source data are provided with this paper.

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

## Acknowledgements
This work was funded by NIH (1U19AI158080, MM, MK). We thank David Weiss, Jake Choby and Dan Andersson for sharing *Escherichia coli*, *Enterobacter cloacae* and *Klebsiella pneumoniae* isolates; Sarah Satola for sharing *Pseudomonas aeruginosa* isolates; Philip Rather and Ralph Isberg for sharing *Acinetobacter baumannii* strains.

## Author contributions
M.M. and M.K. conceived the study. M.M. designed and carried out the experiments. M.K. secured funding and provided resources. M.K. and MM wrote the manuscript. All authors read and approved the manuscript.

## Competing interests
The authors declare no competing interests.
