## [Transparent Peer Review file · Nature Communications]

High-Resolution, High-throughput Detection of Hidden Antibiotic Resistance with the Dilution-and-Delay (DnD) Susceptibility Assay

Corresponding Author: Dr Minsu Kim

Version 0:

Reviewer comments:

Reviewer #1

(Remarks to the Author)

Ma and Kim report on antibiotic susceptibility testing with the Dilution and Delay (DnD) assay.

Overall, the concept is very interesting and the study is well performed.

Currently, heteroresistance, persistence is not detected well with the methods in use, in particular with semi-automated susceptibility testing systems.

Therefore, DnD could improve our understanding of these phenomenonons in the treatment of infections.

I have only few suggestions to further improve the paper:

- The manuscript would profit from adding the clinical perspective: When are heteroresistance and persistence typically observed, when is it important to early detect them (e.g. this is probably not the case in urinary tract infections that are rapidly cleared despite resistant subpopulations).

Additionally, phenotypic heterogeneity is more common in some species (e.g. Mycobacteria, Enterobacter cloacae & others).

- Do we have enough data when DnD (or other method) would be useful for the correct treatment and prognosis? Is there any animal data to support the clinical importance/improved treatment?

If not, this should be discussed in the limitations.

Minor comments:

REF 20 and 21 should be corrected

- Laboratories operate with tight budgets and often a lack of personnel, therefore automated methods are very common.

Could the authors calculate the additional costs for DnD susceptibility testing if 15-20 antibiotics are tested (which is the case for most infections).

Reviewer #2

(Remarks to the Author)

Detailed and argued review in attached file

The manuscript submitted by Muqing Ma and Minsu Kim introduces the Dilution-and-Delay (DnD) assay, a novel high-throughput method for detecting and quantifying antibiotic heteroresistance - rare subpopulations of resistant bacteria that escape detection by standard Minimum Inhibitory Concentration (MIC) testing. By integrating two orthogonal quantification principles into a standard 96-well microplate format, the DnD assay achieves the sensitivity of Population Analysis Profiling (PAP) with the throughput of broth microdilution. The work provides both a potential mechanistic explanation for "unexplained" treatment failures and a practical tool for clinical and research applications.

8. FINAL RECOMMENDATION

8.1 Recommendation: ACCEPT WITH MINOR REVISIONS

This is a high-quality, technically sound manuscript that introduces a valuable methodological advance to the field of antimicrobial susceptibility testing. The conclusions are well-supported by rigorous experimental evidence, and the work addresses a genuine clinical need.

8.2 Specific Revisions Requested

1. Introduction

o Contrary to ETEST® gradient MIC strips, the disk diffusion method described by Matuschek et al. (2014) does not use the diameter of the inhibition zone to infer the MIC but provide “a reading guide on how to read zone diameters on individual species-agent combinations and zone diameter breakpoints calibrated to the EUCAST clinical MIC breakpoints”.

o Explain the selection of a threshold at 1 in 100 million.

2. Expand Methods—Data Analysis Section:

o Line 519-520, precise what a single colony from -80°C glycerol stock corresponds to.

o For Figure 1, indicate the tested concentration of colistin

o Provide the Rf value of the susceptible strain (AMK105) and the fully resistant strain (AMK104)

o Provide explicit criteria for selecting the exponential growth phase (e.g., "OD between 0.05 and 0.5 with ≥ 5 consecutive time points").

o State acceptance thresholds for curve fits (e.g., " $R^2 > 0.95$ for log-linear regression").

o Add a representative example showing raw kinetic data, the fitted line, and intercept annotation.

o For figure 3.c, provide the percentage of resistant isolates for each antibiotic-species combination

o For figure 4, provide the percentage of clinical isolates with a PF > 0 for each of the five species.

o The protocol for the detection of early stages of evolution is missing. What is the concentration of colistin, what are the incubation conditions, ...? As the MIC for the mutants are $\approx < 2\text{mg.l}^{-1}$, below the resistance breakpoint, they may not be classified as “resistant” but having “reduced susceptibility”

3. Supplementary Materials:

o Provide pseudocode or a flowchart for the full analysis pipeline.

o Include a sample Excel or CSV file showing the analysis steps for one representative sample.

o Consider releasing an R or Python script on GitHub for transparency and reproducibility.

o Add MIC values of the clinical isolates in Supplementary Table 1

4. Discussion—Limitations:

o Add a paragraph addressing potential confounds:

Post-antibiotic effects and their impact on Delay-based back-extrapolation.

Antibiotic stability during extended incubations.

Edge cases (e.g., slow-growing resistant variants with severe fitness costs).

o Discuss how users should verify results (e.g., "We recommend confirming Delay-based RF estimates with the Dilution method for routine use").

o Regulatory Implications: Explicitly discuss the challenge of integrating quantitative resistance frequencies into current binary (S/R) reporting structures. Acknowledge that new breakpoints will likely be needed to interpret DnD results clinically.

5. Discussion—Future Directions:

o Explicitly state the need for prospective clinical outcome studies to link DnD-detected heteroresistance to treatment failure rates.

o Suggest mechanistic follow-up (genomic/transcriptomic characterization of heteroresistant subpopulations).

8.3 Why Publication is Warranted

• Methodological Innovation: The DnD assay combines two orthogonal principles in a novel way, achieving a favorable balance of sensitivity, throughput, and practicality.

• Clinical Impact: Heteroresistance is increasingly recognized as a barrier to successful antibiotic therapy. A scalable method to detect it fills a genuine diagnostic gap.

• Robustness: The validation against defined mixtures and equivalence to PAP provide strong evidence for technical soundness.

• Accessibility: Unlike single-cell microfluidics or specialized equipment, DnD uses standard lab infrastructure, maximizing adoption potential.

10. CONCLUSION

This manuscript presents a significant methodological contribution to clinical microbiology. The DnD assay is scientifically rigorous, clinically relevant, and immediately applicable in research and diagnostic settings. With minor revisions to enhance reproducibility and discuss limitations, this work is well-suited for publication and will likely become a standard tool for heteroresistance detection in the field.

Reviewer #3

(Remarks to the Author)

Response to Reviewers

Reviewer #1

We thank the reviewer for the positive assessment of the conceptual advance and technical rigor of this work. We appreciate the emphasis on clinical relevance and agree that clarifying the appropriate clinical contexts and limitations of DnD will further strengthen the manuscript.

Comment 1

The manuscript would profit from adding the clinical perspective: when heteroresistance and persistence are typically observed, and when early detection is important. Phenotypic heterogeneity is also more common in some species.

Response:

We agree and have addressed this comment by adding a clinical perspective to the Discussion. In particular, we now note that heteroresistance and persistence are often recognized retrospectively—when standard susceptibility testing indicates apparent susceptibility, yet antibiotic therapy fails to clear infection (e.g., delayed clearance, breakthrough growth, or relapse). We further emphasize why early detection is clinically important. Finally, we added a brief statement that our high-throughput implementation of DnD suggests organism- and drug-dependent variation in the prevalence of heteroresistance and persistence; we explicitly frame this as a hypothesis generated by our dataset that will require systematic validation in future clinical studies. This discussion has been added to Discussion, lines 427 to 435.

Comment 2

Do we have enough data to show when DnD would be useful for correct treatment and prognosis? Is there animal data? If not, this should be discussed as a limitation.

Response:

We acknowledge this important point. At present, our study does not include prospective clinical outcome data or animal models linking DnD-detected heteroresistance directly to treatment efficacy or prognosis. We have therefore explicitly discussed this as a limitation in the Discussion. We additionally clarified that what is still unclear in the field is how to best treat

heteroresistant and persistent infections. Yet the important first step to develop optimal treatment strategy is to reliably identify and quantify these types of infections, which DnD addresses. This discussion is added to Discussion, lines 436 to 453.

Minor comments

Comment 3

REF 20 and 21 should be corrected.

Response:

Thank you for pointing this out. The two references have been revised in the manuscript to:

20. Clinical and Laboratory Standards Institute. Performance Standards for Antimicrobial Susceptibility Testing, 35th edn. CLSI supplement M100 (Clinical and Laboratory Standards Institute, Wayne, PA, 2025).

21. European Committee on Antimicrobial Susceptibility Testing. Determination of minimum inhibitory concentrations (MICs) of antibacterial agents by broth dilution. *Clin. Microbiol. Infect.* **9**, ix–xv (2003). <https://doi.org/10.1046/j.1469-0691.2003.00790.x>

Comment 4

Laboratories operate with tight budgets and often a lack of personnel. Could the authors estimate additional costs for DnD if 15–20 antibiotics are tested?

Response:

We appreciate the reviewer's question regarding cost considerations when testing multiple antibiotics. DnD is designed to run on standard 96-well microtiter plates using routine consumables. The incremental costs are dominated by plasticware (plates and tips), media, and antibiotics. Importantly, because DnD operates in microliter-scale volumes, media and antibiotic consumption per condition is reduced by an order of magnitude relative to agar-based methods such as PAP. Using standard pricing, this corresponds to a cost on the order of a few dollars to low tens of dollars per isolate per 15–20-drug panel, with the exact value varying by supplier and

assay layout. This discussion has been added to the Discussion section (Discussion, lines 398 to 403).

Reviewer #2 and #3

We thank the reviewer for the detailed, constructive review. We were encouraged by the recommendation of acceptance with minor revisions. Below we address each specific request.

1. Introduction

Comment 1a

Clarify disk diffusion versus ETEST® methodology.

Response:

Thanks for pointing this out. We have revised the text to clarify the methodological distinction between disk diffusion and E-test assays, including how inhibition zones and antibiotic gradients are used to derive susceptibility readouts. The revised text now follows the EUCAST framework as described by Matuschek et al. (2014) (Introduction, lines 69 to 72).

Comment 1b

Explain the selection of the threshold at 1 in 100 million.

Response:

We have clarified the rationale for selecting the 1 in 100 million threshold by emphasizing that it reflects the practical sampling and observation limits of the DnD assay, rather than a biological cutoff. We further note that this frequency scale captures the lowest-frequency range in the operational definition of heteroresistance. This explanation has been added to the Discussion (Discussion, lines 348 to 351).

2. Methods—Data Analysis

Comment 2a

Line 519-520, precise what a single colony from -80°C glycerol stock corresponds to.

Response:

Following your suggestion, we have clarified what a single colony from a $-80\text{ }^{\circ}\text{C}$ glycerol stock corresponds to (Methods, lines 573 to 574).

Comment 2b

For Figure 1, indicate the tested concentration of colistin.

Response:

Thank you for your suggestion. We have added the tested colistin concentration for Fig. 1 (Fig. 1 caption, line 478).

Comment 2c

Provide the R_f value of the susceptible strain (AMK105) and the fully resistant strain (AMK104)

Response:

Based on your suggestion, we have stated RF values for the fully susceptible (AMK105) and fully resistant (AMK104) strains and made necessary revisions (Fig. 1 caption, lines 476 to 480).

Comment 2d

Provide explicit criteria for selecting the exponential growth phase (e.g., "OD between 0.05 and 0.5 with ≥ 5 consecutive time points").

Response:

Following your suggestion, we have explicitly defined objective criteria for selecting exponential growth phases (Methods, lines 608 to 611).

Comment 2e

State acceptance thresholds for curve fits (e.g., " $R^2 > 0.95$ for log-linear regression").

Response:

Thanks for your suggestion. We have added the acceptance thresholds for curve fitting ($R^2 > 0.97$) (Methods, lines 611 to 613).

Comment 2f

Add a representative example showing raw kinetic data, the fitted line, and intercept annotation.

Response:

Based on your suggestion, we have added a representative example showing raw OD₆₀₀ data, fitted line, and intercept annotation (the 11th sheet of Source data Excel file). This is then mentioned and used as a quantitative example in the newly added full analysis flowchart (Supplementary Fig. 8).

Comment 2g

For figure 3.c, provide the percentage of resistant isolates for each antibiotic-species combination.

Response:

Thank you for your suggestion. The percentage of resistant isolates for each antibiotic-species combination has been provided in the 3rd sheet of Source data Excel file.

Comment 2h

For figure 4, provide the percentage of clinical isolates with a PF > 0 for each of the five species.

Response:

Thank you for your suggestion. The percentage of clinical isolates with a PF > 0 for each of the five species has been provided in the 4th sheet of Source data Excel file.

Comment 2i

The protocol for the detection of early stages of evolution is missing. What is the concentration of colistin, what are the incubation conditions, ...? As the MIC for the mutants are $\leq 2\text{mg.l}^{-1}$, below the resistance breakpoint, they may not be classified as “resistant” but having “reduced susceptibility”

Response:

Following your suggestion, we have added full protocol details for the evolution experiments, including colistin concentrations and incubation conditions (Methods, lines 653 to 668), and used the suggested “reduced susceptibility” terminology (Results, Line 257).

3. Supplementary Materials

Comment 3a

Provide pseudocode or a flowchart for the full analysis pipeline.

Response:

Thanks for this helpful suggestion. To clarify the full analysis workflow, we have added a schematic flowchart illustrating the complete data processing and analysis pipeline. This flowchart is now included as Supplementary Fig. 8, and outlines each major step from raw experimental measurements through data normalization, model fitting, and final output metrics. We believe this addition improves transparency and reproducibility of the analysis.

Comment 3b

Include a sample Excel or CSV file showing the analysis steps for one representative sample.

Response:

We appreciate this suggestion. To provide a concrete example of the analysis workflow, we have included a step-by-step implementation for one representative sample in the Source Data Excel file (Sheet 11). This sheet shows the raw input values, intermediate calculations, and final outputs used in the analysis, thereby illustrating how the full pipeline is applied to an individual dataset.

Comment 3c

Consider releasing an R or Python script on GitHub for transparency and reproducibility.

Response:

We appreciate this suggestion. At present, the analysis workflow is provided in Supplementary Fig. 8 to maximize accessibility and ease of use for experimental and clinical laboratories. We did not develop a computer program or script for this research, although we do believe it could facilitate the analysis.

Comment 3d

Add MIC values of the clinical isolates in Supplementary Table 1

Response:

Thank you for your suggestion. We have added the MIC values of the clinical isolates to the 3rd sheet of Source data Excel file. Because of the large number of isolates and antibiotics analyzed, including these values directly in Supplementary Table 1 would substantially increase its size and reduce readability. We therefore provide the complete MIC dataset in the Source Data to ensure transparency while maintaining a concise Supplementary Table.

4. Discussion—Limitations

Comment 4a

Add a paragraph addressing potential confounds:

- *Post-antibiotic effects and their impact on Delay-based back-extrapolation.*
- *Antibiotic stability during extended incubations.*
- *Edge cases (e.g., slow-growing resistant variants with severe fitness costs).*

Response:

Following your suggestion, we have added a paragraph about these factors, including post-antibiotic effects, antibiotic stability during extended incubation, and edge cases such as slow-growing resistant variants. This section has been added to the Discussion (lines 404 to 411).

Comment 4b

Discuss how users should verify results (e.g., "We recommend confirming Delay-based RF estimates with the Dilution method for routine use").

Response:

Following your suggestion, we now recommend cross-validating delay-based RF estimates with dilution-based measurements for routine use. See Discussion (line 411 to 416).

Comment 4c

Regulatory Implications: Explicitly discuss the challenge of integrating quantitative resistance frequencies into current binary (S/R) reporting structures. Acknowledge that new breakpoints will likely be needed to interpret DnD results clinically.

Response:

Thank you for your suggestion. We have expanded the Discussion to explicitly address the regulatory implications of DnD. We now discuss the challenge of integrating quantitative resistance-frequency measurements into the current binary (S/R) susceptibility reporting framework and note that interpreting such outputs will require new interpretive schemes, potentially including an additional category (e.g., heteroresistant) rather than direct replacement of existing MIC-based breakpoints. We further emphasize that outcome-linked clinical validation and consensus standardization will be required to establish how DnD results should be used in routine clinical reporting. See Discussion (lines 436 to 453).

5. Discussion—Future Directions

Comment 5a

Explicitly state the need for prospective clinical outcome studies to link DnD-detected heteroresistance to treatment failure rates.

Response:

We appreciate this suggestion. We have explicitly addressed this point in the Discussion by stating that, while DnD enables sensitive quantification of heteroresistant subpopulations, linking DnD-detected heteroresistance to treatment failure rates will require prospective clinical outcome studies. We emphasize that such outcome-linked validation is a critical next step to determine how resistance-frequency measurements should inform clinical decision-making. Please see Discussion (Lines 436 to 453).

Comment 5b

Suggest mechanistic follow-up (genomic/transcriptomic characterization of heteroresistant subpopulations).

Response:

We agree with the reviewer that mechanistic follow-up studies will be important for elucidating

the origins of heteroresistant subpopulations. We have therefore expanded the Discussion (Lines 424 to 426) to explicitly note that genomic and transcriptomic characterization of DnD-identified resistant minorities represents a natural next step to link quantitative resistance-frequency measurements with underlying molecular mechanisms.